# Aptamer BC 007’s Affinity to Specific and Less-Specific Anti-SARS-CoV-2 Neutralizing Antibodies

**DOI:** 10.3390/v13050932

**Published:** 2021-05-18

**Authors:** Annekathrin Haberland, Oxana Krylova, Heike Nikolenko, Peter Göttel, Andre Dallmann, Johannes Müller, Hardy Weisshoff

**Affiliations:** 1Berlin Cures GmbH, Robert-Rössle-Str. 10 Blg. D79, 13125 Berlin, Germany; 2Leibniz-Forschungsinstitut für Molekulare Pharmakologie im Forschungsverbund Berlin e.V. (FMP), Robert-Rössle-Str. 10, 13125 Berlin, Germany; Krylova@fmp-berlin.de (O.K.); Nikolenko@fmp-berlin.de (H.N.); 3Berlin Cures GmbH, Knesebeckstr. 59-61, 10719 Berlin, Germany; goettel@berlincures.de (P.G.); Jmueller@berlincures.de (J.M.); 4Department of Chemistry, NMR Facility, Humboldt University of Berlin, Brook-Taylor-Straße 2, 12489 Berlin, Germany; andre.dallmann@chemie.hu-berlin.de (A.D.); weiss@chemie.hu-berlin.de (H.W.)

**Keywords:** aptamer, autoantibody, BC 007, COVID-19, re-purposing, SARS-CoV-2 antibody

## Abstract

COVID-19 is a pandemic respiratory disease that is caused by the highly infectious severe acute respiratory syndrome coronavirus 2 (SARS-CoV-2). Anti-SARS-CoV-2 antibodies are essential weapons that a patient with COVID-19 has to combat the disease. When now repurposing a drug, namely an aptamer that interacts with SARS-CoV-2 proteins for COVID-19 treatment (BC 007), which is, however, a neutralizer of pathogenic autoantibodies in its original indication, the possibility of also binding and neutralizing anti-SARS-CoV-2 antibodies must be considered. Here, the highly specific virus-neutralizing antibodies have to be distinguished from the ones that also show cross-reactivity to tissues. The last-mentioned could be the origin of the widely reported SARS-CoV-2-induced autoimmunity, which should also become a target of therapy. We, therefore, used enzyme-linked immunosorbent assay (ELISA) technology to assess the binding of well-characterized publicly accessible anti-SARS-CoV-2 antibodies (CV07-209 and CV07-270) with BC 007. Nuclear magnetic resonance spectroscopy, isothermal calorimetric titration, and circular dichroism spectroscopy were additionally used to test the binding of BC 007 to DNA-binding sequence segments of these antibodies. BC 007 did not bind to the highly specific neutralizing anti-SARS-CoV-2 antibody but did bind to the less specific one. This, however, was a lot less compared to an autoantibody of its original indication (14.2%, range 11.0–21.5%). It was also interesting to see that the less-specific anti-SARS-CoV-2 antibody also showed a high background signal in the ELISA (binding on NeutrAvidin-coated or activated but noncoated plastic plate). These initial experiments suggest that the risk of binding and neutralizing highly specific anti-SARS CoV-2 antibodies by BC 007 should be low.

## 1. Introduction

Aptamer BC 007, currently under development for the neutralization of functionally active autoantibodies in autoantibody-associated heart failure, was recently reported to be a specific binder of DNA-binding-susceptible sequence sections of proteins crucial for the growth of SARS-CoV-2, such as the spike protein and the RNA-dependent RNA polymerase [1]. This would provide the opportunity to quickly check for in vivo efficiency in COVID-19 patients given the advanced state of this drug’s development. BC 007 is currently in phase 2a of clinical testing in heart failure patients. In phase 1 of clinical testing, BC 007 showed excellent safety and tolerability profiles [2].

On the way to the clinical trial for COVID-19 effectivity testing, there is one important question. This is whether BC 007 would interfere with anti-SARS-CoV-2 antibodies (anti-SARS-CoV-2 ABs) the patients develop while their bodies are fighting SARS-CoV-2.

The reason for this potential risk is simple: in its current indication, BC 007 neutralizes pathogenic autoantibodies, namely functionally active autoantibodies against G-protein-coupled receptors (_f_GPCR-AABs) [2,3,4], as they occur in heart failure patients [5]. Even though highly different in the variable region, antibodies against SARS-CoV-2 and _f_GPCR-AABs are the same type of molecule. This explains, therefore, the potential risk of BC 007 interfering with SARS-CoV-2 ABs in COVID-19 patients while being used for virus neutralization. This is especially true in light of the fact that BC 007 neutralizes an entire class of autoantibodies [3].

However, it must also be considered that the involvement of cross-reactive anti-SARS-CoV-2 antibodies in severe and long-lasting disease courses (Long-COVID) is increasingly being discussed [6,7,8,9,10,11,12,13]. These different types of anti-SARS-CoV-2 ABs have to be distinguished. Even the occurrence of functionally active autoantibodies against G-protein-coupled receptors in patients suffering from Long-COVID symptoms has already been reported [14].

We, therefore, searched for publicly accessible sequences of specific and cross-reactive patients’ anti-SARS-CoV-2 ABs to check for BC 007 binding. We already knew from former investigations with this aptamer (originally selected as a potential short-lasting thrombin inhibitor for transient anticoagulation during coronary bypass graft surgery [15] but failed in this indication because of a lack of a persistent effect which resulted in a suboptimal dosing profile (too high a dose for a therapeutic effect)) that a section of the protein binding sequence is already enough to show the binding.

With respect to neutralizing anti-SARS-CoV-2 AB proteins, many publications can be found [16,17,18,19,20,21,22,23,24]. However, only a few publications exist in which the antibodies were investigated with respect to virus specificity and cross-reactivity [17,21], and only one of those has the sequences accessible [21]. Drawing on this previous publication, we focused on two sequences of SARS-CoV-2 neutralizing antibodies that do not show cross-reactivity and one that showed cross-reactivity with murine tissues. We checked the intermolecular interaction of DNA-binding susceptible sequence sections of these antibodies with BC 007 using nuclear magnetic resonance spectroscopy (NMR), isothermal calorimetric titration (ITC), and circular dichroism spectroscopy (CD). For verification of the results, the binding of the entire monoclonal antibodies onto immobilized aptamer BC 007 was tested exemplarily using CV07-209 and CV07-270 as the specific and nonspecific antibodies, respectively.

## 2. Materials and Methods

### 2.1. Materials

Aptamer BC 007 is a 15mer ssDNA of the following sequence: 5′-GGTTGGTGTGGTTGG-3′. It was synthesized by BioSpring GmbH (Frankfurt, Germany) under GMP conditions. 5′-biotinylated BC 007 was synthesized by BioTez GmbH (Berlin, Germany). Peptides were synthesized by Biosyntan GmbH (Berlin-Buch, Germany) using solid-phase synthesis with a purity of >95% (MALDITOF; MALDI AXIMA Assurance, Shimadzu, Japan).

Monoclonal antibodies CV07-209, CV07-270, and 011-138 were kindly supplied by J. Kreye, S.M. Reincke, and H. Prüss [21].

### 2.2. Nuclear Magnetic Resonance (NMR) Spectroscopy

NMR measurements were carried out as recently described by Weisshoff et al. while investigating the interaction of BC 007 with sequence sections of growth-essential SARS-CoV-2 proteins [1]. While investigating the interaction of BC 007 with sequence sections of anti-SARS-CoV-2 mABs, all NMR measurements were carried out on a Bruker AV600 spectrometer (Bruker Biospin, Rheinstetten, Germany) at 600 MHz in 90/10 H_2_O/D_2_O at 298 K. The final concentration of BC 007 and the peptides (ARDGVIPPRFDY, ARARGSSGWYRIGTRWGNWFDP, AGSDNYGFPYNGMDV) was 1 mM.

The Watergate w5 pulse sequence included in the Bruker pulse program zggpw5 was used to suppress the solvent signal. The following acquisition parameters were used: time domain, 65 K; number of scans, 512; sweep width, 24 ppm; 90° high power pulse, 13.8 µs.

### 2.3. High-Sensitivity Isothermal Titration Calorimetry (ITC)

ITC was used to further investigate one of the peptide sequence-sections, ARDGVIPPRFDY, with BC 007, which showed somewhat ambiguous results in NMR. The experiments were carried out on a MicroCal PEAQ-ITC microcalorimeter (Malvern Panalytical GmbH, Germany) at 25 °C. BC 007 and the peptides were solved in 50 mM sodium phosphate and 150 mM NaCl buffer at pH 7.0.

In an initial screening experiment, a solution of 4 mM peptide was titrated in 2 µL steps into the 200 µM BC 007 solution in the calorimeter cell, in order to be comparable to results published by Weisshoff et al. [1]. The generated data were used to adjust the necessary concentration range to allow for precise measurements, which amounted to 12 mM peptide in the syringe and 600 µM BC 007 in the measuring cell. The time intervals between the injections were adjusted to 200 s, sufficient for the heat signal to return to baseline. The reaction mixtures were stirred at 750 rpm. Dilution heats that were associated with the addition of peptides to the buffer (separate control experiments) had small constant and negligible values. The instrument software (MicroCal PEAQ-ITC Analysis) was used for data analysis.

### 2.4. Circular Dichroism (CD) Spectroscopy

The CD spectra of solutions of 21.5 µM BC 007 without (control) and with peptides of different concentrations in water were recorded at 25 °C using a Jasco J-720 spectrometer. For temperature control, a CD-Peltier Element Jasco PTC-423S/15 (JASCO, Japan) was utilized. Spectra were recorded at wavelengths between 200 and 350 nm under the following conditions: cell length, 0.2 cm; scanning speed, 100 nm/min; response time, 1 s; band width, 1 nm; accumulation, 5; data pitch, 0.1 nm. D-10-(+)-Camphorsulfonic acid was used for CD spectrometer calibration at 290 nm.

The CD spectroscopy data measured the differential absorption of left and right circularly polarized light. Data are reported as ellipticity (expressed in millidegrees (mdeg)). The corresponding control spectra (blank) were subtracted.

### 2.5. Enzyme-Linked Immunosorbent Assay (ELISA) Binding Experiments

ELISA technology was used to test the binding of monoclonal antibodies CV07-209, CV07-270, and 011-138 for control onto immobilized BC 007. For this purpose, ELISA plates (F96, Thermo Scientific, Cat no. 163746) were precoated with 0.05 µM NeutrAvidin (Thermo Scientific, Cat no. F 144746) in carbonate coating buffer pH 9.25 overnight at 4 °C. After washing using the washing buffer (20 mM Tris-HCl, 150 mM NaCl, 4 mM KCl, and 0.1% Tween 20, pH 7.2), 5′-biotin-labeled BC 007 (0.1 µM) was added to the wells, which served for the estimation of the specific binding. Completely uncoated wells (only incubated and activated with carbonate coating buffer) or wells coated with NeutrAvidin served for the estimation of the unspecific binding (control wells).

The BC 007 specified wells and also all control wells were, after washing, blocked with 1% BSA/PBS/0.05% Tween 20; washed again and loaded with the monoclonal antibodies in a concentration range between 1.5 and 50 nM diluted in dilution buffer that consisted of 20 mM Tris-HCl, 150 mM NaCl, 4 mM KCl, and 0.05% Tween 20, pH 7.2. Afterwards, the plate was decanted and washed, and the secondary detection antibody affiniPure goat anti-human IgG (H+L)-POD (Cat. no. 109-035-003, Dianova, Germany) at a dilution of 1:10,000 was applied. For detection and quantification of bound monoclonal antibodies, the TMB/hydrogen peroxide color reaction was used. The measurement was performed on an Anthos ht II reader at the measuring wavelength of 450 nm and reference wavelength of 620 nm. The experiments were repeated twice (n = 3), all run in duplicate. The absolute and unmodified raw data were compared. In addition, the percentage values of the individual experiments and clones (highest value set to 100%) were compared.

The comparable affinity of the secondary antibody to all 3 mABs (011-138, CV07-209, and CV07-270) was tested before.

## 3. Results and Discussion

Very advantageous for the processing of this question about the anti-SARS-CoV-2 binding of BC 007 was the detailed characterization of monoclonal human anti-SARS-CoV-2 ABs [21]. While these authors achieved their goal of identifying an anti-SARS-CoV-2 AB that could be the basis for the development of a therapeutic anti-SARS-CoV-2 mAB, Kreye et al. [21] isolated and investigated hundreds of single clones from 10 different donors (COVID-19 patients). Of those, about 40 proved to be strongly neutralizing mABs. They then selected the 18 best mABs for further characterization.

From all the clones, CV07-209 was the best for its purpose. It was the best not only with respect to binding and neutralizing SARS-CoV-2 in vivo in the COVID-19 hamster model but also with respect to not showing cross-reactivity with different murine tissues. An additional two of those anti-SARS-CoV-2 AB clones were used for a more detailed investigation of their binding with the spike protein of SARS-CoV-2 (CV07-250: PDB ID: 6XKQ and CV07-270: PDB ID: 6XKP). Of these two, the first, CV07-250, was a SARS-CoV-2-specific mAB that showed good binding to the receptor-binding domain (RBD) of the spike protein and neutralization with no cross-reactivity to murine tissues, while the second, CV07-270, the one that showed the least effect with respect to RBD-ACE2 binding competition, did show cross-reactivity with smooth muscle tissue (off-target binding) in their experiments [21] (Table 1).

These are all different specific characteristics that make these three mABs valuable candidates for an investigation of their potential interactions with BC 007, given the publicly available sequences of the CDR-H3 and L3 regions of all of their investigated clones and the entire heavy- and light-chain sequences for CV07-250 and CV07-270 (PDB IDs 6XKQ and 6XKP, respectively) [21]. We checked those sequences for theoretical susceptibility and accessibility to interact with DNA according to previous work [1].

Luscombe and Thornton identified favored amino acid base pairs after they investigated 129 protein-DNA families. With respect to BC 007, these would be arginine, lysine, histidine, and serine with guanine generating stronger hydrogen bonds and proline and phenylalanine with thymine generating weaker van der Waals contacts [25]. This approach was transferred to the antibodies.

Identified sequence sections were investigated in a second step for interaction with BC 007 using NMR spectroscopy. However, none of the published sequences allowed one to assume pronounced DNA-binding susceptibility. The CDR-H3 sequences of CV07-270 and CV07-209 (ARARGSSGWYR and ARDGVIPPR, respectively) showed the slight possibility of binding to BC 007 through their arginine sidechains. Therefore, the two sequences described in most detail (CV07-209: ARDGVIPPRFDY, theoretical pI value of 6.0, ExPASy [26] and CV07-270: ARARGSSGWYRIGTRWGNWFDP, theoretical pI value of 11.54) were further investigated for binding with BC 007 using NMR spectroscopy. AGSDNYGFPYNGMDV (CDR-H3 from CV07-250, theoretical pI value of 3.56) was included in order to add a sequence of another cross-reactivity-free antibody (Figure 1A).

Even though the light chain of CV07-250 was reported to be heavily involved in the binding on the RBD sequence of the spike protein of SARS-CoV-2, the entire CDR-H3, as usual, was observed to participate. Therefore, possible binding of BC 007 on CDR-H3 should definitely interfere with the binding of the anti-SARS-CoV-2 antibody to its target, thus weakening adaptive immunity in case of application for therapeutic purposes. Our NMR investigation, however, did not show any interaction of BC 007 with the CDR-H3 sequence of the neutralizing anti-SARS-CoV-2 ABs of the patients, which does not show affinity for other tissues (Figure 1A). The peptide spectra of the mixture of BC 007 with sequences from the specific CV07-209, which did not show any binding (signal shifts) (Figure 1B), proved that the small observed binding when looking at the quadruple formation in the range between 11.5 and 12.5 ppm (Figure 1A) can in fact be neglected.

This was re-checked using isothermal titration calorimetry (ITC) and circular dicroism spectroscopy (CD) (Figure 2). ITC excellently confirmed the NMR data and revealed a binding constant for ARDGVIPPRFDY, the sequence section of the specific neutralizing mAB (which showed only very small imino signals in the NMR), in the millimolar range. The data were just outside the ITC possible measuring range that would enable an evaluation of a specific molecular interaction.

With CV07-270, the CDR-3 of the heavy chain was, as usual, the dominant binding sequence on the spike protein of SARS-CoV-2 [21]. With this anti-SARS-CoV-2 AB, the CDR-H3 region (ARARGSSGWYRIGTRWGNWFDP) was shown to have the highest chance of interacting with DNA. Since this sequence is easily aggregated and precipitated, it was only measured at molar ratios which kept it in solution of 1:1 and 1:2 (BC 007/peptide) in the NMR and CD, respectively. BC 007 has the great advantage of showing its intermolecular interactions (binding) in a specifically formed quadruplex structure, traceable not only via NMR but also via CD spectroscopy [1]. The NMR imino proton signals of the quadruplex structure in the range between 11.5 and 12.5 ppm are this way a correlate of the folding success. Electrostatic interactions alone as a cause of binding can be excluded since it had been shown before that peptides showing even higher charge did not interact with BC 007 [1]. CD spectroscopy also confirmed this binding, as seen in Figure 2. Here, the binding of BC 007 with ARARGSSGWYRIGTRWGNWFDP, ARDGVIPPRFDY, and AGSDNYGFPYNGMDV is shown and compared.

With the CD measurements, it became visible that ARARGSSGWYRIGTRWGNWFDP (CV07-270, PDB ID 6XKP) showed interaction with BC 007, forcing the DNA molecule into its quadruplex structure, while with ARDGVIPPRFDY, the molecule that showed only very slight interaction in the NMR, only an extremely weak interaction was visible. As already expected from the NMR, no interaction was detectable with AGSDNYGFPYNGMDV (CV07-250, PDB ID: 6XKQ). Even a molar excess of peptide of 2.3 times with respect to the BC 007 concentration did not result in any binding.

These binding affinities could have been confirmed by investigating the binding of the full antibodies onto immobilized BC 007 in the ELISA format (Figure 3). Only CV07-209 and CV07-270 were available. However, these are the antibodies that show the greatest differences in terms of specificity. Moreover, a monoclonal antibody that showed affinity for the beta1-adrenoceptor in the bioassay was added for comparison. Here, it was clear that this anti-beta1-receptor antibody showed the highest affinity for immobilized BC 007 (Figure 3(C1)) followed by CV07-270 (Figure 3(B1)). CV07-209, however, did not show affinity for BC 007 (Figure 3(A1)). In addition, the specific binding of the individual clones was shown in comparison to their nonspecific binding as a percentage of the highest value (=100%) in order to bring out the details of the single clones more clearly (Figure 3(A2–C2)). The statistics made for the raw data and the percentage values are given in Table 2. With respect to the statistical difference between specific and nonspecific binding with CV07-209, be aware that this was all below the nonspecific binding onto NeutrAvidin-coated plates. All absolute values are close to the baseline.

Additionally, it is remarkable that the antibody that showed self-reactivity before (CV07-270) also showed, in percentage terms, the highest nonspecific binding onto plastic material and NeutrAvidin-coated wells.

It might, therefore, be a perfect candidate for initiating the autoimmune process in a COVID-19 patient, as often observed [6,7,9,10,11] and even discussed to be a reason for disease severity [12].

If this observed interaction of BC 007 with the CDR-H3 sequence of this antibody could provide future therapeutic options for treating patients suffering from autoimmune-associated post-COVID-19 symptoms, as exemplarily described by Masuccio et al. [8], this should be further investigated in more detail. For sure it will interact with functionally active autoantibodies against G-protein-coupled receptors. Functionally active autoantibodies against G-protein-coupled receptors have already been found in many Long-COVID patients suffering from a variety of different neurological and/or cardiological diseases [14].

In conclusion, BC 007 did not interfere with the CDR regions of the neutralizing anti-SARS-CoV-2 ABs of patients, which were specific and did not show cross-reactivity, while BC 007 showed binding with the one which showed strong cross-reactivity to murine tissues [21]. This might mean that the risk of interference with essential anti-SARS-CoV-2 ABs is low in this case, while there could even be the possibility of neutralizing self-reactive anti-SARS-CoV-2 ABs that cross-react with other tissues. One has to keep in mind that these are just examples. However, so far, only very few sequences of well-characterized anti-SARS-CoV-2 ABs are available. Nevertheless, they offered a great chance to start such investigations. Other sequences of published highly specific neutralizing anti-SARS-antibodies such as REGN10933 (PDB ID 6XDG) [22], LY-CoV555 (PDB ID 7KMG) [23], and IgG1-kappa 2E8 Fab (PDB ID 12E8) [24] were checked theoretically for possible binding with BC 007. Reassuringly, no susceptible sequence segments could be detected.

## Figures and Tables

**Figure 1 viruses-13-00932-f001:**
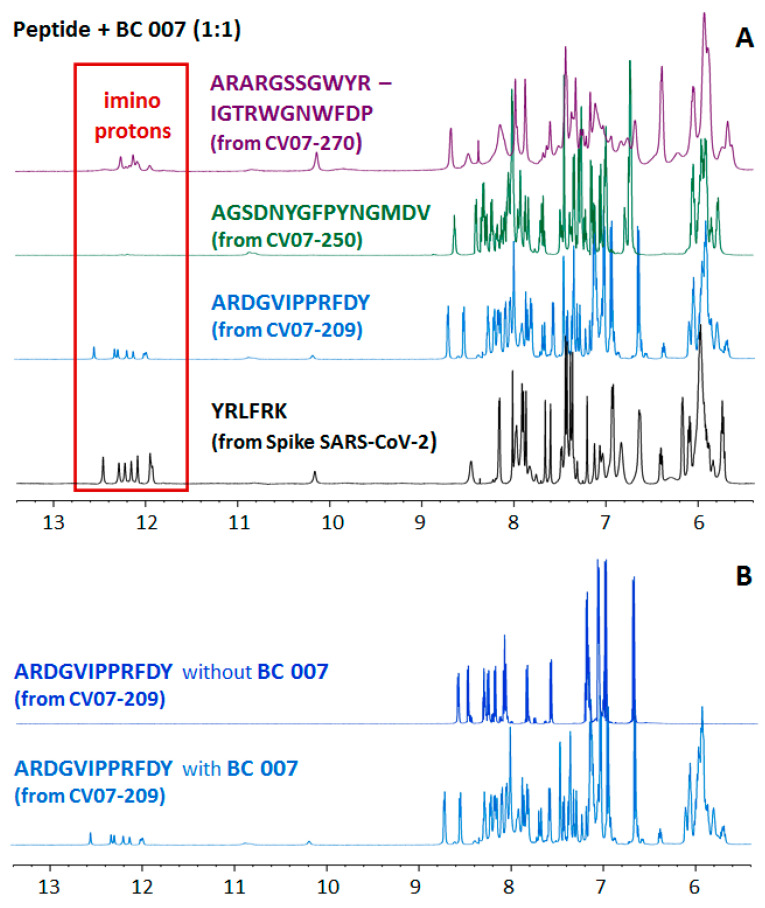
NMR spectroscopic evaluation of the molecular interaction of BC 007 with published CDR-H3 sequences of the anti-SARS-CoV-2 mABs of patients. (**A**) Upper spectrum (purple): The upper NMR spectrum of BC 007 (1 mM) in combination with ARARGSSGWYRIGTRWGNWFDP (published CDR-H3 of CV07-270 [21]) (1:1) shows BC 007-folding-specific imino signals in the range of 11.5–12.5 ppm. Second from top (green): The spectrum of 1 mM BC 007 in the presence of 1 mM AGSDNYGFPYNGMDV (published CDR-H3 of CV07-250 [21]) does not show any folding of BC 007, excluding any binding or molecular interactions. Third from top (blue): Here, 1 mM ARDGVIPPRFDY (published CDR-H3 of CV07-209 [21]) showed a very small amount of quadruplex-specific imino signals between 11.5 and 12.5 ppm. Bottom spectrum (black): For comparison of the visualization of the quadruplex signal as binding and folding success: SARS-CoV-2 spike protein occurring sequence YRLFRK-caused fold of BC 007 as discussed before in detail by Weisshoff et al. [1]. (**B**) Upper spectrum: Peptide sequence of the published CDR-H3 region of CV07-209 (ARDGVIPPRFDY, 1 mM) alone. Lower spectrum: Here, 1 mM ARDGVIPPRFDY in the presence of 1 mM BC 007 does not show significant shifts in the aromatic protons of Phe and Tyr. Charge-, pH-, and ionic strength-dependent amide protons of all amino acids are showing only small changes in the chemical shift but not in the coupling constants, again excluding any strong interaction of the CDR-H3 region of CV07-209 with BC 007.

**Figure 2 viruses-13-00932-f002:**
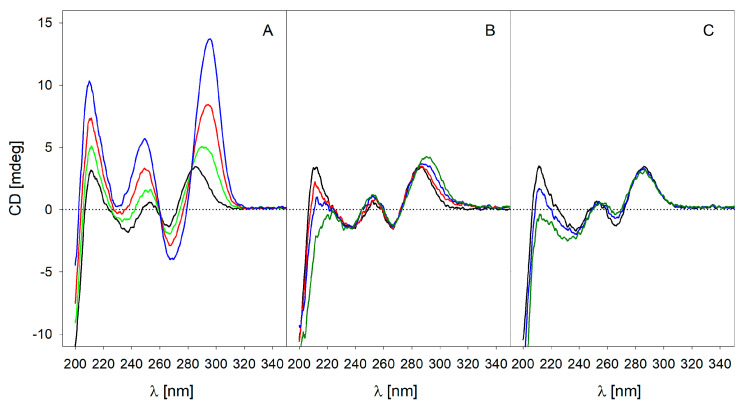
CD spectroscopic evaluation of the molecular interaction of BC 007 with sequence sections of the CDR-H3 regions of anti-SARS-CoV-2 antibodies. The BC 007 (21.5 µM) folding success in presence of no peptides (control, black curves) and increasing concentrations of peptides (**A**) ARARGSSGWYRIGTRWGNWFDP (source: CV07-270, PDB ID 6XKP), (**B**) ARDGVIPPRFDY (source: CV07-209 of [21]) and (**C**) AGSDNYGFPYNGMDV (source: CV-250, PDB ID: 6XKQ) up to a molar excess as indicated by the color of the curve was measured. Green, red, and blue curves show the BC 007 spectra with additional presence of 6.25, 12.6, and 25.0 µM peptide, respectively.

**Figure 3 viruses-13-00932-f003:**
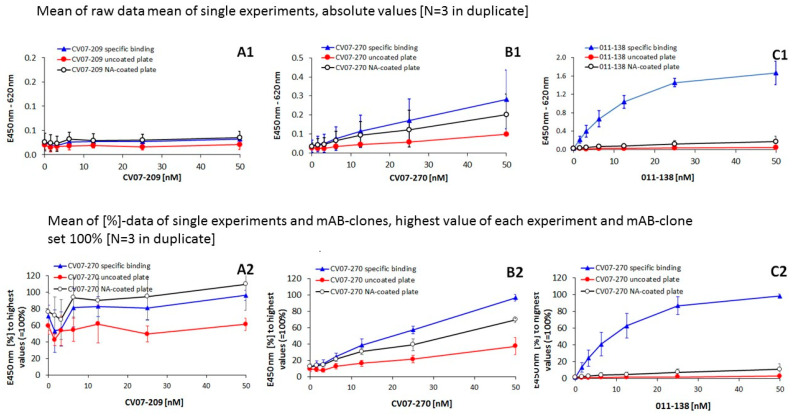
Binding of monoclonal antibodies onto immobilized BC 007. Upper row presents raw data of (**A1**) neutralizing anti-SARS-CoV-2 antibody CV07-209 (highly specific) and (**B1**) neutralizing SARS-CoV-2 antibody CV07-270 (showing also cross-reactivity [21]) in comparison to (**C1**) a BC 007-affine monoclonal antibody that activates beta1-adrenoceptors (clone 011-138, origin: Kreye et al., unpublished). Lower row data are in percent, with the highest value of each experiment and clone set to 100%: (**A2**) CV07-209, (**B2**) CV07-270, and (**C2**) 011-138. Shown in each case is the specific binding onto immobilized BC 007 (blue line), the nonspecific binding onto immobilized NeutrAvidin (the carrier for BC 007–biotin, control, black line), and the nonspecific binding of the mABs onto the carbonate buffer activated but noncoated plastic plate (control, red line). Note the different axis scales with the raw data (**A1**,**B1**,**C1**), which were chosen to show details clearly.

**Table 1 viruses-13-00932-t001:** Overview over the published and used anti-SARS-CoV-2 AB sequence sections.

Clone Name	Cross-Reactive [Reference]	PDB No.	Sections of CDR-H3	Binding of Sequence Sections Onto BC 007
CV07-250	no [21]	6XKQ	AGSDNYGFPYNGMDV	no
CV07-209	no [21]	n.a.	ARDGVIPPRFDY	no
CV07-270	yes [21]	6XKP	ARARGSSGWYRIGTRWGNWFDP	yes

**Table 2 viruses-13-00932-t002:** Statistical evaluation of the binding of the single mAB clones onto immobilized BC 007 compared to control binding (NA-coated plate and uncoated activated plastic plate); values from Figure 3.

Concentration mAB Clone		50 (nM)	25 (nM)	12.5 (nM)	6.25 (nM)	3.125 (nM)	1.56 (nM)
Condition vs. Condition							
**CV07-209**							
Figure 3(A1)	specif/uncoated	n.s.	*	n.s.	n.s.	n.s.	n.s.
	uncoated/NA-coated	n.s.	*	n.s.	n.s.	n.s.	n.s.
Figure 3(A2)	specif/uncoated	***	**	n.s.	*	n.s.	n.s.
	uncoated/NA-coated	*	**	n.s.	*	n.s.	*
**CV07-270**							
Figure 3(B1)	specif/uncoated	*	*	*	n.s	n.s	n.s
	uncoated/NA-coated	*	n.s.	n.s.	n.s.	n.s.	n.s.
Figure 3(B2)	specif/uncoated	***	***	***	***	**	**
	uncoated/NA-coated	***	**	***	***	***	**
**011-138**							
Figure 3(C1)	specif/uncoated	***	***	***	***	***	***
	uncoated/NA-coated	**	**	*	*	*	n.s.
Figure 3(C2)	specif/uncoated	***	***	***	***	***	**
	uncoated/NA-coated		*	*	*	n.s.	n.s.

Student’s *t*-test, 2-sided, type 2; *p* values < 0.05 (*) considered significant and *p* values < 0.01 (**) and < 0.001 (***) considered highly significant; n.s. = not significant.

## Data Availability

The data presented in this study are available in the article.

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
