# Peer review of "Aptamer BC 007’s Affinity to Specific and Less-Specific Anti-SARS-CoV-2 Neutralizing Antibodies"

_viruses, 2021, doi:10.3390/v13050932_

Round 1
Reviewer 1 Report
Thanks for giving the opportunity for reviewing this article. I see authors have manage to improve the overall presentation of the MS. Regarding my initial comments, I notice also they made an effort to provide an answer. I will support the publication and I hope following minor comments can be included:
Minor points:
1) I have noticed the technical discussion with other reviewers in the cover letter and I will appreciate if the author could include a SM and add any relevant information for reproduction of their results.
2) In addition, as the research in COVID-19 is moving faster than any other field in sciences nowadays, I would ask to consider following citations about the structure, dynamics and stability of the spike (S) protein (https://doi.org/10.3390/ma13235362 and https://doi.org/10.1039/D0NR03969A) which is key viral protein in cell entry and cellular recognition to the ACE2 receptor.
Author Response
Dear Reviewer,
Thank you very much for reviewing the revision and for your suggestions.
1) I have noticed the technical discussion with other reviewers in the cover letter and I will appreciate if the author could include a SM and add any relevant information for reproduction of their results.
Authors: We have not added any additional material. It is a "Brief report" with only the results we present in the paper. We very much hope that this can be acceptable.
2) In addition, as the research in COVID-19 is moving faster than any other field in sciences nowadays, I would ask to consider following citations about the structure, dynamics and stability of the spike (S) protein (https://doi.org/10.3390/ma13235362 and https://doi.org/10.1039/D0NR03969A) which is key viral protein in cell entry and cellular recognition to the ACE2 receptor.
Authors: Dear reviewer, thank you very much for pointing out these fascinating and important papers. We read them carefully and noticed that the sequence section of the Spike protein we investigated in [1] is not a “hot spot” for stability or instability as investigated by Moreira and team.
However, since aptamers combine several different targets and thus modes of action, they are worth testing for their utility to patients. However, in this submitted manuscript, we describe binding to more or less specific anti-SARS antibodies. So we do not see the connection of our manuscript with the issue you raised, which is indeed very important. So we decided to not include these two references [2][3] in this manuscript. If we will work with the spike protein again, we will certainly take these results into account.
We hope very much that this will be fine with you.
References
- Weisshoff, H.; Krylova, O.; Nikolenko, H.; Düngen, H.-D.; Dallmann, A.; Becker, S.; Göttel, P.; Müller, J.; Haberland, A. Aptamer BC 007 - Efficient Binder of Spreading-Crucial SARS-CoV-2 Proteins. Heliyon 2020, 6, doi:10.1016/j.heliyon.2020.e05421.
- Moreira, R.A.; Guzman, H.V.; Boopathi, S.; Baker, J.L.; Poma, A.B. Characterization of Structural and Energetic Differences between Conformations of the SARS-CoV-2 Spike Protein. Materials 2020, 13, 5362, doi:10.3390/ma13235362.
- Moreira, R.A.; Chwastyk, M.; Baker, J.L.; Guzman, H.V.; Poma, A.B. Quantitative Determination of Mechanical Stability in the Novel Coronavirus Spike Protein. Nanoscale 2020, 12, 16409–16413, doi:10.1039/D0NR03969A.
Reviewer 2 Report
The authors have made many changes and improvements to the manuscript writing and data presentation that in my opinion improve the paper significantly.
I still think that some minor edits to English language throughout is needed but the title and abstract are now much better, as is most of the text explanations.
Author Response
Dear reviewer,
We had the manuscript read and corrected again by a native speaker.
For this purpose, we downloaded the manuscript and accepted all changes, before we now added the corrections (tracked change version of the manuscript) which are:
Line 47-48: “a patient develops while their body combates SARS” replaced by: “the patients develop while their bodies are fighting SARS-CoV-2.”
Line 70: “many publication are to be found” replaced by: “many publications can be found”
Line 95: erased the “now”
Line 123: "served" replaced by: “was utilized”
Line 148: “exploited” replaced by: “used”
Line 165: “show” replaced by “showing”
Line 187: changed the order of the words; instead of: “were in a second step investigated” now: “were investigated in a second step”
Line 207: “could have indeed been neglected” replaced by: “can in fact be neglected”
Line 220: “to show” replaced by: “of showing”
Line 226: “as to be seen” replaced by “as seen”
Line 232: “slim” replaced by “slight”
Line 267: added: this for: “this should be further…”
We very much hope that this has improved the English.
This manuscript is a resubmission of an earlier submission. The following is a list of the peer review reports and author responses from that submission.
Round 1
Reviewer 1 Report
The work by Haberland et al. reports a very short result based on two main figures of the aptamer BC 007 against SARS-CoV-2. Also, it has been shown no effect on the common SARS-CoV-2 antibodies. In general, the MS is not so transparent. At this moment, I fail to see a well-organised manuscript with several mistakes in style, data presentation and message. Before going to a second revision I would ask the authors to address very crucial concerns.
Major:
1) Quality of figures are very low. Mistakes are often spot over main text and figure. For instance sequence name in line 175 ARARGSSGWYRIGTRWGNWFDP does not agree with Figure 2. The same Figure 2 does not show clear message. It can be reorganised.
2) The title does not show a conventional style. Even it does not sound well in English
3) Abstract, please do not cite PDB ids (use the name of the antibodies, CV07-250, CV07-2780). Please add missing word aptamer for BC 007. The style is not correct, excessive use of short sentences.
4) Line 37. BC 007 has not been introduce, what is the chemical structure, properties, and current use of it.
5) Line 40: what is the current state of the clinical phase regarding BC 007? Add message. 6) Line 44-46. Please rephrasethe message.
7) Line 50: replace GPCR-fAABS by fGPCR-AAbs
8) Line 52-62: Please rewrite the whole text not clear the message.
9) Pay attention to methods section. I am afraid this part has been already used by authors in their previous publication (10.1016/j.heliyon.2020.e05421). I suggest to avoid copy-paste if they really want to insist in a fair revision. See below:
2.1 Materials (here) ---> the same in 10.1016/j.heliyon.2020.e05421
2.2 Nuclear magnetic resonance (NMR) spectroscopy 70% common with 10.1016/j.heliyon.2020.e05421
2.3 and 2.4. Present the same problems.
10) Line 110-111, Problem with English: ... while investigating anti-SARS-CoV-2 ABs in great detail [6]. While reaching
11) Line 174, Problem with English:
- ... also confirmed this binding (Figure 2). Figure 2 shows and
- Break apart the results and conclusion: At this moment it is very difficult to assess it.
I will possible accept to assess again the article under resubmission, but I suggest to use word doc virus-journal template. As it will improve the assessment and it will make the article easy to read.
Reviewer 2 Report
This study could have been very interesting and worthwhile. Unfortunately in the present form it is not possible to fully evaluate and make conclusions based on the writing, content and data presented.
The tile is really not well worded and is confusing, as is the abstract which is entirely the incorrect format - it should begin with SARS-CoV-2 background explanations including immunity. The results and conclusions are not clearly stated and it is not even explained what BC007 is. It is not until the first line of the introduction that this molecule is described and even then it is confusing. Eventually it appears to be an aptamer selected to bind paratope of auto antibodies. The hypothesis is actually sound, would BC007 interact with SARS-CoV-2 antibodies shown to be multi reactive. But the data and presentation with explanations that go to support this are not possible to fully interpret and conclude the meaning from. A fatal flaw is the use of peptides that correspond to CDR3 regions of antibodies. This not only assumes they will fold correctly as in antibodies but ignore contributions of the entire antibody paratope in binding the aptamer. The figures presented are lacking controls - should use the auto antibody that BC007 was designed to bind. Related peptides or mutants should be used to control. Whole Ab binding studies must be done. The CDRs are not defined in context of the whole antibody and it becomes confusing to understand the antibody nomenclature, function and sequence without a table summarising all the data. In the text figure 2 has estimated binding constants but these are not apparent. Line 320 folding is the wrong word, line 325 "YRLFRK-caused fold of BC007" is confusing.
The final conclusion from lines 203-207 does not reflect the study at all and seems randomly added. It also contains a second example of the use of the word adoptive/adopted when surely the authors are referring to adaptive immunity.
If the authors have used proof-reading-service.com they may wish to revise whether or not this has improved the writing at all. I would suggest a native English speaker would be required to fix the writing throughout.